# Multivariable Air-Quality Prediction and Modelling via Hybrid Machine Learning: A Case Study for Craiova, Romania

**DOI:** 10.3390/s24051532

**Published:** 2024-02-27

**Authors:** Youness El Mghouchi, Mihaela Tinca Udristioiu, Hasan Yildizhan

**Affiliations:** 1Department of Energetics, ENSAM, Moulay Ismail University, Meknes 50050, Morocco; y.elmghouchi@umi.ac.ma; 2Department of Physics, Faculty of Science, University of Craiova, 13 A.I. Cuza Street, 200585 Craiova, Romania; 3Engineering Faculty, Energy Systems Engineering, Adana Alparslan Türkeş Science and Technology University, Adana 46278, Turkey; hyildizhan@atu.edu.tr

**Keywords:** air pollution, hybrid machine learning, low-cost sensors, PM sensor, urban monitoring

## Abstract

Inadequate air quality has adverse impacts on human well-being and contributes to the progression of climate change, leading to fluctuations in temperature. Therefore, gaining a localized comprehension of the interplay between climate variations and air pollution holds great significance in alleviating the health repercussions of air pollution. This study uses a holistic approach to make air quality predictions and multivariate modelling. It investigates the associations between meteorological factors, encompassing temperature, relative humidity, air pressure, and three particulate matter concentrations (PM10, PM2.5, and PM1), and the correlation between PM concentrations and noise levels, volatile organic compounds, and carbon dioxide emissions. Five hybrid machine learning models were employed to predict PM concentrations and then the Air Quality Index (AQI). Twelve PM sensors evenly distributed in Craiova City, Romania, provided the dataset for five months (22 September 2021–17 February 2022). The sensors transmitted data each minute. The prediction accuracy of the models was evaluated and the results revealed that, in general, the coefficient of determination (R^2^) values exceeded 0.96 (interval of confidence is 0.95) and, in most instances, approached 0.99. Relative humidity emerged as the least influential variable on PM concentrations, while the most accurate predictions were achieved by combining pressure with temperature. PM10 (less than 10 µm in diameter) concentrations exhibited a notable correlation with PM2.5 (less than 2.5 µm in diameter) concentrations and a moderate correlation with PM1 (less than 1 µm in diameter). Nevertheless, other findings indicated that PM concentrations were not strongly related to NOISE, CO_2_, and VOC, and these last variables should be combined with another meteorological variable to enhance the prediction accuracy. Ultimately, this study established novel relationships for predicting PM concentrations and AQI based on the most effective combinations of predictor variables identified.

## 1. Introduction

Air pollution has gained significant attention as a prominent research topic due to its substantial implications for public health and the environment [1]. On a global scale, exposure to PM is responsible for 3% of cardiopulmonary-related deaths and 5% of lung-cancer-related fatalities, as the World Health Organization (WHO) reported in 2013 [2]. Short-term exposure from hours to days to high concentrations of PM10 has been observed to affect respiratory health adversely. However, it is essential to note that long-term exposure over months to years to PM2.5 carries a higher health risk than PM10. Extended exposure to PM2.5 has been linked to increased mortality rates due to respiratory issues [3,4,5], heart diseases [6,7,8], lung cancer [3], and stroke [9]. On average, PM2.5 reduces the population’s life expectancy by 8.6 months, as reported by the WHO in 2013 [2]. During the COVID-19 pandemic, PM2.5 emerged as one of the most significant pollutant agents contributing to increased death rates associated with COVID-19 [10]. Some studies have even suggested that PM1 mainly affects male residents in urban areas, who face a higher risk of lung cancer incidence [3]. Vulnerable groups to air pollution exposure include children, the elderly, and individuals with chronic illnesses [11]. Furthermore, low- and middle-income communities tend to bear a more significant burden of exposure to elevated PM concentrations than wealthier communities [12].

The alert threshold recommended by WHO in 2021 for PM2.5 is 15 µg/m^3^ for a 24 h mean and 5 µg/m^3^ annual mean. The daily limit recommended for PM10 is 45 µg/m^3^, and the annual limit is 15 µg/m^3^. Another important detail is that it should not exist for more than 3–4 exceedance days per year [1]. The European Union air quality standards are 25 µg/m^3^ for PM2.5 and 40 µg/m^3^ for PM10 (one-year average). The European Environment Agency declared in November 2023 that Europe had 253,000 premature deaths in 2021 from chronic exposure to fine PM. Moreover, the World Quality Report 2021 emphasized that only 3.4% of 6735 monitored cities met the standards in 2021.

The expansion of urban areas has made a notable contribution to the deterioration of environmental quality, primarily owing to the dust generated by construction sites and the development of transportation infrastructure. Given that transportation networks are vital for a city’s economic progress, governmental bodies are faced with the imperative task of seeking strategies to redirect a portion of road traffic through bypass routes. Additionally, there is a concerning trend of diminishing green spaces in favor of urban expansion.

Table 1 presents a list of abbreviations and nomenclature. Some units are added where necessary.

In the realm of the literature, the application of Machine Learning (ML) models, also referred to as data-driven models, for the modelling, prediction, and forecasting of air quality, with a focus on atmospheric components such as PM1, PM2.5, and PM10 concentrations, has been explored to a limited extent. Researchers have combined various ML techniques with Feature Selection (FS) methods to identify the most relevant predictor variables and enhance prediction accuracy. This collectively sheds light on applying ML techniques to air quality prediction and forecasting and underlines the significance of FS and hybrid modelling approaches in achieving accurate predictions.

Table 2 provides an overview of the state-of-the-art air quality prediction and forecasting methods, encompassing hybrid and non-hybrid FS-ML models over the past five years. It includes a concise description of each method, the objective function it addresses, the location and source of the data used, the predictor variables incorporated, the time-series resolution (e.g., minute, hourly, and daily average), and the strengths and limitations associated with each model.

For instance, in reference [13], the authors employed three distinct ML models—Mixture Discriminant Analysis (MDA), Bagged Classification and Regression Trees (Bagged CART), and Random Forest (RF)—for predicting PM10 hazards in Barcelona, Spain. Simulated Annealing (SA) was applied as an FS technique to reduce the data dimension and select appropriate predictor variables. The results showed accuracies exceeding 87% and precisions surpassing 86% for all three ML models.

In [14], the focus was on accurately predicting PM2.5 concentrations. The authors introduced a hybrid model comprising a deterministic prediction module and a Random Fourier Extreme Learning Machine (RF-ELM), combined with an interval prediction module. This approach effectively provided concentration intervals based on upper and lower bounds derived during the deterministic prediction phase.

In [15], the researchers examined the impact of anthropogenic emissions and meteorological factors on PM2.5 concentrations in Hubei Province, China, using a random forest model in conjunction with a meteorological normalization method. The findings indicated that anthropogenic emissions increased PM2.5 concentrations by approximately 33.3%, while meteorological conditions contributed to an 8.8% increase.

In [16], considering climate-influencing factors, the authors proposed an intelligent hybrid air-quality-forecasting system. This system incorporated an FS technique (relief-F algorithm), a multi-objective optimization algorithm (MOCBO), and a modified fuzzy neural network. The Air Quality Index (AQI) was computed based on concentrations of several air pollutants, including PM2.5, PM10, SO_2_, CO, NO_2_, and O_3_. The results demonstrated that this proposed system outperformed eleven comparison models, which included two ML models (general neural networks—ELM and deep learning neural networks—LSTM) combined with five FS techniques and three multi-objective optimization algorithms (MODA, MOPSO, and MOBO).

**Table 2 sensors-24-01532-t002:** The state of the art on air quality forecasting research for the last five years.

Ref.	A Brief Description	Objective Function	Data Location and Source	Predictors	Time-Series	Strengths	Limitation
[17]	An automated air quality forecasting system is developed for daily forecasts based on five various ML models: MLR, MLP, RF, GBDT, and SVR, combined with an FS technique.	PM2.5, PM10, SO_2_, NO_2_, O_3_, CO	Seven cities in China: Beijing, Shanghai, Guangzhou, Chengdu, Xi’an, Wuhan, and Changchun. (http://www.cnemc.cn) (accessed on 15 January 2022).	Daily pressure, 2 m temperature, relative humidity, precipitation, visibility, and total cloud cover. (http://data.cma.cn) (accessed on 15 January 2022).	Daily average	Development of an automated air quality forecasting system based on five various ML models.	Feature importance scores were calculated by the RF model, in which the predictor variables were checked individually.
[14]	Hybrid model based on a deterministic prediction module (RF-ELM) combined with an interval prediction module.	PM2.5	Three major cities in China are Guang Zhou, Shenzhen, and Zhuhai.	----	Daily average	The use of an interval prediction module.	These are very complex models.
[15]	RF model combined with a meteorological normalization method.	PM2.5	Hubei Province, China. https://quotsoft.net/air/ (accessed on 15 January 2022).	Included 2 m temperature, 2 m dewpoint temperature, 10 m u-component of wind, 10 m v-component of wind, surface pressure, total precipitation, boundary layer height, and downward surface solar radiation.	Hourly	The use of a meteorological normalization method.	Only a quantification of air pollution was performed. No forecasting and/or modelling was made.
[16]	Hybrid air quality forecasting system based on relief-F algorithm combined with a MOCBO and a modified fuzzy neural network.	AQI	Shanghai, Hangzhou, and Nanjing are three regions with severe air pollution in China.	PM2.5, PM10, SO_2_, CO, NO_2_, and O_3_ concentrations, average temperature (°C), cumulative precipitation (CP, mm), average wind speed (AWS, m/s), and average relative humidity.	Daily average	A comparison with other ML models and FS methods.	One combination of inputs was found for AQI forecasting.
[13]	MDA, Bagged CART, and RF combined with SA.	PM10	A total of 75 stations over Barcelona, Spain.	Minimum temperature, maximum temperature, normalized difference vegetation index, precipitation, wind speed, wind direction, elevation, road density, topographic wetness index, land use, terrain roughness index, distance from water body, land use, and lithology.	Annually average	The use of many FS-ML models and comparison with others.	One combination of inputs was found for PM10 forecasting.
[18]	An ANN model was used to forecast daily pollutant concentrations. Real-time correlation (RTC) was applied to improve the quality of the forecasts.	PM10, PM2.5, NO_2_, and O_3_	A total of 32 continuous air-quality-monitoring stations in Delhi, India.	CAVG_DAY0 CAVG_DAYM1 BLH_DAYN T2M_DAYN RH_DAYN IS975_DAYN IS950_DAYN IS925_DAYN U10_DAYNM1_DAYN V10_DAYNM1_DAYN TP_DAYN FIRE_DAYNM3_DAYNM1.	Daily average	Application of Real-Time Correction (RTC) technique.	ANN is a stochastic method, which means that one cannot obtian the same results for the same dataset. No FS was applied.
[19]	A hybrid early-warning artificial intelligence framework (ICEEMDAN-OS-ELM) was proposed.	PM2.5, PM10, and lower atmospheric visibility	Gladstone, Brisbane, Mackay Region, Newcastle, and Sydney, Australia.	---	Hourly	The results are benchmarked with many ML models.	The main common weakness is that one should have data (measures) for obtaining data (forecasts).
[20]	Forecasting AQI using a long short-term memory (LSTM) neural network model combined with a variational mode decomposition (VMD) and a sample entropy.	AQI	Beijing and Baoding, China. https://www.aqistudy.cn/historydata/ (accessed on 15 January 2022).	---	Daily average	A comparison with other models was performed.	No FS was applied.
[21]	Air pollutant concentration forecasting was performed by combining an EWT decomposition algorithm with MAEGA and NARX neural networks.	PM2.5, SO_2_, NO_2_, CO	Beijing in China.	---	---	A comparison was made with the VMD-MAEGA-NARX, EWT-MAEGA-SVM, MAEGA-NARX, EWT-NARX, and EWT-ARIMA-NARX models.	No inputs and no FS were applied.
[22]	A dynamic multiple equation (DME) model (a linear model).	PM2.5	Santiago, Chile.	Temperature, wind speed, relative humidity, wind direction, and CO.	Hourly and daily average	A comparison with SARI-MAX and ANN models.	Complex model structure.

In all these studies, the researchers delved into various applications involving hybrid and non-hybrid ML models for predicting AQI and/or other pollutant concentrations. A common practice among these studies was integrating meteorological factors and anthropogenic emissions into their models. However, these studies generally adhered to a single set of predictor variables, sometimes employing FS techniques and sometimes not, in their analyses. What sets them apart is that none systematically compared all conceivable combinations of predictor variables to discern the optimal ones regarding their correlations, relationships, and approximations to the stated objective function.

In contrast to the prior research outlined in Table 2, the current study aims to identify the most robust correlations by adopting a holistic approach considering all possible variable combinations. This innovative research endeavors to assess atmospheric air pollution using more advanced hybrid software. The key novelty and primary objectives of this study are as follows:i.Implementing an Autonomous Anomaly Detection method during data preprocessing to identify and exclude anomalous data points.ii.Identifying spatial and temporal hazards detected by the study’s sensors/stations.iii.Clustering and decomposing data based on the significance of AQI in terms of health implications.iv.Analyzing partial dependence and estimating the importance of each predictor variable considered.v.Determining the optimal combinations of predictor variables for predicting AQI and other related pollutant concentrations through a comprehensive FS approach.vi.Evaluating the performance of five hybrid FS-ML models for predicting a one-minute series of PM10, PM2.5, and PM1 and then AQI.vii.Developing new physical models for estimating PM10, PM2.5, PM1, and AQI.viii.Creating a new interface module to provide PM10, PM2.5, PM1, and AQI predictions based on the provided predictor variables.

Incorporating all relevant factors into the process of air pollution prediction is crucial for the accurate detection and assessment of air quality. This study aims to assess the added value of hybrid FS-ML models in air pollution prediction. The primary objective of this research is to examine the impact of three meteorological parameters—T, P, RH, noise levels, and carbon dioxide emissions—on the PMs and AQI.

In the context of air quality monitoring, this study holds significance for the following aspects:i.Analyzing pollution episodes in Craiova in line with World Health Organization (WHO) recommendations.ii.Evaluating the correlations between meteorological parameters, AQI, and PM concentrations and interrelations among different PM fractions, such as PM1, PM2.5, and PM10.iii.Investigating the influences of noise and carbon dioxide (CO_2_) on PM concentrations.

This study aims to better understand the complex interplay between meteorological factors and air quality, contributing to more accurate and insightful air pollution predictions.

## 2. Data and Statistical Analysis

### 2.1. Local Weather Information

The study was conducted in Craiova City (Figure 1), the capital of Dolj County and the sixth-largest city in Romania by population number. According to the National Institute of Statistics data and the 2022 census, Craiova has 243,765 inhabitants. The distribution of the population by age category in Craiova city is as follows: 12.6% young population (0–14 years), 61% adult population (15–60 years), and 16.4% elderly population (>60 years). The city has a surface area of 81 km^2^ and is in continuous development. Craiova is in the Oltenia Plain, near the east bank of the Jiu River. The climate is temperate continental with some Mediterranean influences, having long, hot summers and short, mild winters. As a feature, five heat islands formed in paved areas and surrounded by buildings have been identified in Craiova [23]. From an economic point of view, in 2022, the SW region was placed in the sixth rank of Romania’s eight administrative regions, with a GDP per capita equal to 58 pps (Eurostat). PM10 sources at the local level were identified as fixed sources (industry and fossil fuel power stations) that produce 86.54 t/year, surface sources (slag and ash deposits, vegetation fires, waste incineration, construction sites, demolition, and infrastructure works) with a contribution of 59.1 t/year, and mobile sources (road and air traffic) producing 0.48 t/year [23]. 

The dataset used in this paper was provided by twelve monitoring PM sensors (16000207, 16000208, 16000209, 1600020A, 1600020B, 1600020C, 1600020D, 1600020E, 1600020F, 16000238, 1600023A, and 820002C3), which are evenly distributed over the entire surface of Craiova, at a 100 m altitude (Figure 2). Eleven sensors are the PM Smoggie model, and one is the A3 model (820002C3). The mentioned sensors are part of an independent network of sensors, different from the official one. Smoggie PM provides PM concentrations (1 µg/m^3^ resolution, ±5% accuracy, and R^2^ = 0.99%, 81.6%, and 99.9% for all fractions’ coefficient of correlation to reference gravimetric sampler) and three meteorological parameters like air temperature (0.5 °C resolution and ±1 °C accuracy), relative humidity (1% resolution and ±2% accuracy), and pressure (±0.25% accuracy) at a higher spatial and temporal resolution. In addition, A3 can track volatile organic compounds (±5% accuracy), formaldehyde (10 ppb resolution and ±5% accuracy), ozone (10 ppb resolution and ±5% accuracy), carbon dioxide (1 ppm resolution and ±5% accuracy), and noise level (1 dB resolution and ±10% accuracy). 

The National Research and Development Institute for Industrial Ecology (INCD-ECOIND), Romania, and the Observatoire de la qualité de l’air en Île-de-France (AIRPARIF), France (from the EU) tested the A3 and Smoggie PM sensors in laboratory chambers conditions (under known aerosol concentrations, controlled temperature of 20 °C, and relative humidity conditions of 50%). Both laboratories stated that the checked sensors met the variability conditions, and the correlation coefficients between the sensors and the reference were good and very good. To verify the accuracy of the measurements of the sensors, the results were analyzed using the Pearson statistical correlation method and compared with the results given by the reference instruments. Another important detail is that, after the sensors are produced, they are introduced by the manufacturer into a particular chamber and compared with a reference sensor. The differences between the devices and the reference are calculated. The corrections are included in the equipment’s software for automated systems like A3 and Smoggie PM sensors, (according to the recommendations made by the mentioned EU-accredited laboratories). The sensors indicate the corrected values, and the trueness error is compensated. All sensors used in this study were in their first year of life.

These twelve sensors are part of an independent sensor network built during a volunteering project for educational purposes. The sensors are in different high schools and public institutions in Craiova, with one exception: a sensor located in a residential area. Each high school “adopted a sensor” during an awareness campaign about the importance of clean air for health. The sensors are evenly distributed in Craiova over its entire surface area. Power or Wi-Fi failures can occur in high schools. For this reason, some sensors recorded less data. The sensors work properly, but the dataset is incomplete for short intervals for some sensors. Before starting everything, the dataset must be analyzed using an Autonomous Anomaly Detection method. All sensors were produced by a Romanian start-up focused on innovation and were calibrated by the manufacturer. Two international independent laboratories stated that the PM Smoggie and A3 sensors under-evaluate PM concentrations.

The official network has only four stations in Craiova and six in Dolj County. The development of the independent network of PM sensors came about due to the lack of measures taken by local authorities during air pollution episodes. Laser scattering is the method used by the used sensors to measure PM concentrations. The official stations measure PM concentrations using the gravimetric method. According to the EU regulations, the method used by the official stations from National Environment Agencies is the gravimetric method. This method is based on the weight differences of filters pre- and post-sampling. Regardless of the methods used, there are correlations between their results. Both methods (laser scattering and gravimetric) are good, but each has its limitations. 

The measurements started on 22 September 2021 and ended on 17 February 2022. The PM Smoggie sensors measure three meteorological parameters (T, P, and RH) and three particulate matter concentrations: PM1, PM2.5, and PM10. The 820002C3 sensor is more complex and, additionally, can measure volatile organic compounds (VOC), noise, CO_2_, formaldehyde, and ozone. All parameters are measured every minute. The locations of the 12 sensors are indicated in Figure 2. 

The data are first analyzed by the Quartile Method as an Autonomous Anomaly Detection method for eliminating or ignoring anomalous data items, and are subdivided into training and validating datasets. The Quartile Method is a statistical approach for identifying outliers in a dataset. It involves calculating the interquartile range and setting thresholds based on this range.

AQI reports the air quality daily, helping people to understand how the local air quality affects their health. To calculate AQI, converting the measurement unit transmitted by the sensor’s parts per million (ppm) into µg/m^3^ is necessary. The limit values of AQI are presented in Table 3 (European Environment Agency), and the computed AQI versus PM concentrations for all 12 stations are illustrated in Figure 3, where it shows how important PM concentrations are in AQI and emphasizes pollution episodes. 

For the AQI computation, the formulas adopted by the US-EPA were applied, in which the AQI ranges from 0 to 500, with 0 meaning a good environment and 500 meaning a hazardous environment. The formulas used here are as given in Equation (1). Then, we used the equation two times (for PM2.5 and PM10). The worst sub-index (the max value) that communicates the AQI is given by the formula.
(1)$${\mathrm{Index}}_{p}=\left[\frac{{\mathrm{IH}}_{i}–{\mathrm{IL}}_{o}\;}{{\mathrm{BPH}}_{i}–{\mathrm{BPL}}_{o}}\right]\left(\mathrm{Cp}–{\mathrm{BPL}}_{o}\right)+{\mathrm{IL}}_{o}$$
where ${\mathrm{Index}}_{p}$ is the index for the pollutant p; Cp is the truncated concentration of the pollutant p; ${\mathrm{BPH}}_{i}$ is the concentration breakpoint, i.e., greater than or equal to Cp; ${\mathrm{BPL}}_{o}$ is the concentration breakpoint, i.e., less than or equal to Cp; ${\mathrm{IH}}_{i}$ is the AQI value related to ${\mathrm{BPH}}_{i}$; and ${\mathrm{IL}}_{o}$ is the AQI value related to ${\mathrm{BPL}}_{o}$.

Figure 4 clusters the computed AQI values for the sensors studied between 22 September 2021 and 17 February 2022. For each sensor, the data are classified as Good, Moderate, Unhealthy for sensitive groups, Unhealthy, Very Unhealthy, or Hazardous clusters (Table 3). The sensor that registered remarkable AQI values for Hazardous was 1600020F (4473 values), between 30 September 2021 at 12:00 and 6 October 2021 at 19:34. The sensor with ID 1600020D registered 1094 Very Unhealthy and 9152 Unhealthy values for AQI. Very Unhealthy values were recorded for 13 October 2021, and 17 November 2021, 25–26 September 2021. Unhealthy values were recorded between 1 October 2021 and 20 November 2021. Other sensors recorded very Unhealthy and Unhealthy AQI values, but they may be overlooked by the 1600020D and 1600020F sensors. The sensor 1600020F is near the airport and a busy entrance of the city. The sensor 1600020D is in a residential area at the city’s outer edge, with many houses whose inhabitants use fossil fuels for house heating. Electricity is not widely used in heating houses because of the price. 

In general, Figure 4 shows that, between 22 September 2021 and 17 February 2022, all sensors provided a one-minute series for PM2.5 and PM10 over the recommended limit. The official network of sensors (www.calitateaer.ro) did not indicate any active alert related to exceeding the PM2.5 and PM10 concentrations. A monitored system might relate to a datalogger device to detect unexpected AQI values and set alerts. Also, there is a need for a device that can track the air mass trajectory between the source and the destination.

Table 4 presents the input and output parameters that will be used further when the performances of the proposed hybrid FS-ML models are evaluated. 

### 2.2. Correlation between the PM1, PM2.5, and PM10 Concentrations

In Figure 5, the correlation between PM1, PM2.5, and PM10 was examined across the 12 studied stations/sensors. The observed correlation ranged from 0.95 to 1, highlighting a robust correlation among the investigated PMs. This outcome suggests a significant interdependence, signifying that a slight alteration in one of the PMs may influence the others. Moreover, it implies the capability to predict one of these PMs with exceptional accuracy and precision based on the provided values of the remaining PMs.

### 2.3. Evaluation Criteria and Statistical Indices

The performance of the proposed models was assessed by a method suggested by Badescu [24], in which a performance score (φ) for a model is defined as:(2)$$\varphi =\mathrm{rank}\left(\mathrm{MBE}\right)+\;\mathrm{rank}\left(\mathrm{RMSE}\right)+\;\mathrm{rank}\left(\mathrm{TS}\right)+\;\mathrm{rank}\left(R^{2}\right)+\;\mathrm{rank}\left(\mathrm{WIA}\right)+\;\mathrm{rank}\left(\mathrm{SBF}\right)$$

Higher values of φ signify a poor model performance. The indicators used in Formula (2) are, respectively, Mean Bias Error (MBE), Root Mean Square Error (RMSE), T-Statistic (TS), Coefficient of Determination (R^2^), Willmott’s Index of Agreement (WIA), and Slope of Best-Fit line (SBF). They are given by Equations (3)–(8):(3)$$\mathrm{MBE}=\frac{1}{K}\sum \left(v_{p}^{i}-v_{m}^{i}\right)$$
(4)$$\mathrm{RMSE}={\left(\frac{1}{K}\sum {\left(v_{p}^{i}-v_{m}^{i}\right)}^{2}\right)}^{\frac{1}{2}}$$
(5)$$\mathrm{TS}={\left[\frac{\left(K-1\right){\mathrm{MBE}}^{2}}{\left({\mathrm{RMSE}}^{2}-{\mathrm{MBE}}^{2}\right)}\right]}^{1/2}$$
(6)$$R^{2}=1-\frac{\sum {\left(v_{p}^{i}-v_{m}^{i}\right)}^{2}}{\sum {\left(v_{m}^{i}-\bar{v_{m}}\right)}^{2}}$$
(7)$$\mathrm{SBF}=\frac{\left[\sum \left(v_{p}^{i}-\bar{H_{p}}\right)\left(v_{m}^{i}-\bar{v_{m}}\right)\right]}{\sum {\left(v_{m}^{i}-\bar{v_{m}}\right)}^{2}}$$
(8)$$\mathrm{WIA}=1-\frac{\left[\sum {\left(v_{p}^{i}-v_{m}^{i}\right)}^{2}\right]}{\sum {\left[\left|v_{p}^{i}-\bar{v_{m}}\right|+\left|v_{m}^{i}-\bar{v_{m}}\right|\right]}^{2}}$$

Another analysis is based on Standard Deviation σ and Mean Absolute Percentage Error (MAPE). These are given by Equations (9) and (10):(9)$$\sigma ={\left[\frac{K\left({\mathrm{RMSE}}^{2}-{\mathrm{MBE}}^{2}\right)}{\left(K-1\right)}\right]}^{1/2}$$
(10)$$\mathrm{MAPE}=\frac{100}{K}\sum \left|\frac{\left(v_{p}^{i}-v_{m}^{i}\right)}{v_{m}^{i}}\right|$$

In these formulas, K represents the total number of measures and $v_{p}^{i}$, $v_{m}^{i}$, and $\bar{v}$ are the ith predicted value, ith measured value, and the mean value of the corresponding output (AQI, PM1, PM2.5, or PM10), respectively.

## 3. Hybrid FS-ML Models

This work employed five different hybrid FS-ML models for predicting and modelling the AQI and PMs concentrations over Craiova. Then, the performance of each model was checked, and the best one was adopted. The ML models employed were Artificial Neural Network (ANN), Support Vector Machine (SVM), Decision Tree (DT), Gaussian process regression (GPR), and Linear Regression (LR). They are briefly described below.

### 3.1. Machine Learning Models

i.Artificial Neural Network

ANN is a stochastic and nonlinear technique inspired by speculating the information processing of brain neurons. An ANN consists of many nodes and their connections. Each node corresponds to a unique function called the ‘activation objective function’. The connection between the nodes represents the weight of the measure operating through, which provides ANN a memory. The output of the ANN is fixed by the weight and the activation objective function [25]. In addition, due to its strong nonlinear affinity potential, ANN has been broadly utilized in many fields. For more information, the readers are referred to the reference.

ii.Support Vector Machine

SVM, originally recommended by [26], is a deterministic method and a generalized classifier that groups data based on supervised learning. SVM is based on finding the support vector to form the optimum taxonomy hyperplane in the training set. Generally, SVM implements a pivot loss function to compute empirical threats by improving its sparsity and strength [27]. 

iii.Decision Tree

Originally announced in [28], DT is a deterministic and supervised learning method. DT indicates the benefits of randomization approaches, alternate analysis, and classifying and grouping techniques. The main significant uses of DT include discovering data anomalies, discovering data patterns, and providing accurate results [29]. Due to its reliability and diversity, DT is one of the most employed ML models for prediction and modelling.

iv.Gaussian Process Regression

Based on Bayesian statistics, GR uses historical data and data-fitting approaches to construct a robust model [30]. An appropriate kernel function can explicitly display the nonlinear relationships between predictors and objective functions. Its average and covariance functions can identify a Gaussian process f(x). Thus, the important point of regression is to make the relationship between predictors and objective function meet: $y_{i}=f\left(x_{i}\right)+\epsilon_{i}$, where the objective function $y_{i}$ differs from the function values f(x) by additive noise ϵ that is supposed to be an independent coefficient.

v.Linear Regression

LR was employed to find a linear equation that can describe the relationship between the predictor variables x_i_ and the response variable y (the objective function) through known data and using a linear equation [31]. The most common form of regression problem is linear regression, by which one should find the line that most closely fits the data provided according to a particular criterion. The relationship between predictors x and objective function y should meet the criterion: y = ax + b.

### 3.2. Feature Selection: Integral Feature Selection Method

Before using the datasets in any ML model, it is necessary to conduct a statistical analysis and the pruning of sizable environmental datasets. In this work, an Integral Feature Method was employed with an ML model to optimize the dataset to be used in the prediction stage. This method, which was published in [32], belongs to Input Variable Selection (IVS) and has been elaborated to provide the best possible combination of predictor variables that can be employed for the prediction, forecasting, and modelling of an objective function. According to this method, the number of possible combinations of inputs can be computed by Equation (11).
(11)$$\mathrm{Comb}=\sum_{p=1}^{n}C_{n}^{p}=\sum_{p=1}^{n}\frac{n!}{\left(n-p\right)!p!}$$
where n is the total number of the predictor variables.

### 3.3. Modelling: Least Square Regression

Like the Gradient Descent method, LSR is based on a line that makes a vertical distance from the data points to the regression line as small as possible. The best line of fit is given as a function that should reduce the sum of squares of the errors [33,34]. LSR has been widely used by researchers worldwide for both regression and modelling problems. In this work, using LSR, new relationships between the considered objective function (AQI, PM1, PM2.5, or PM10) and the best predictor variables were elaborated.

## 4. Methodology

For evaluating the performances of the hybrid FS-ML models studied here, the main steps in our methodology are summarized as follows (Figure 6):Start the algorithm.Import the inputs and outputs data.First, the data are pre-processed by applying normalization and Autonomous Anomaly Detection, are loaded to each studied ML model, and then are subdivided into training (80% of data) and testing (the remaining data).Compute the total number of combinations based on the data size loaded using Equation (11).Start a first loop based on the size of the provided data, K1.Compute the number of combinations for each ith considered size and then start a second loop for each value of K2.Use the combnk(V, K) function for producing a matrix with K columns.Load the ML model, load the data, and compute the considered output parameter.Save the computed values and go to the next iteration.After obtaining the predicted values by all considered combinations, the result is imported by a second algorithm in which the statistical analysis is performed.The best combinations of inputs are found and then the algorithm is ended.

**Figure 6 sensors-24-01532-f006:**
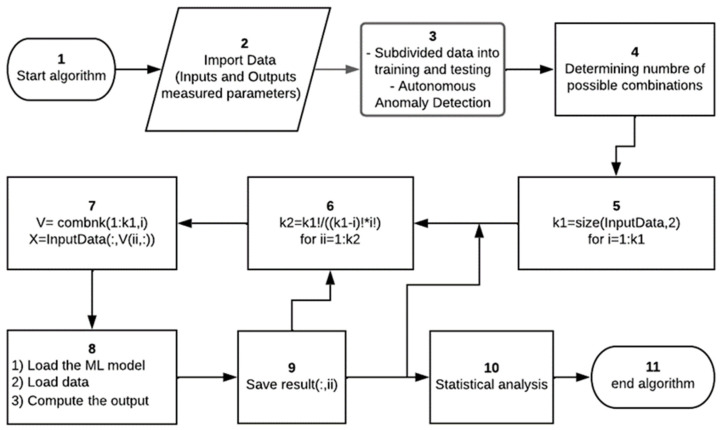
Flowchart for the proposed methodology.

## 5. Results and Discussion

In this study, comprehensive air pollution prediction and modelling were carried out by including many atmospheric variables with a holistic approach. For three input meteorological variables here, there are seven possible combinations. The corresponding values computed for each PM output were stored and statically compared to determine the best combinations to provide the considered PM with the best possible accuracy. 

All combinations can be expressed as:-Comb1: Temperature-Comb2: Pressure-Comb3: Humidity-Comb4: Temperature and Pressure-Comb5: Temperature and Humidity-Comb6: Pressure and Humidity-Comb7: Temperature, Pressure, and Humidity

Before applying the hybrid FS-ML model, the prediction capability of each ML model was checked for predicting PM10 concentrations, and then the best model for each station was chosen. The analysis used a single combination of inputs that included all predictor variables. The ML models were compared and ranked based on their performance score φ and then on their coefficient of determination R^2^ (confidence level is 0.95), MAPE, and σ. These indicators are illustrated as dark blue for the rank, black for R^2^, yellow for MAPE, and light blue for σ. The results of the comparison are shown in Figure 7.

As is clearly shown, the best accuracy was for the DT model. With this model, the predictions were statistically very significant. The corresponding R^2^ was closer to 1, indicating perfect correlation and relationships between the measured and predicted values, whereas other dispersion indicators were closer to zero. More results can be obtained from the same figure. Compared to those presented in Table 2, the correlations found here indicate very accurate predictions and outperformed the results of the models studied and applied by several researchers. For example, in [13], the authors found an accuracy of >87% and a precision of >86% for the hazard prediction of PM10 in Barcelona. Here, the accuracy and precision found by the DT model were close to 98% for almost all stations/sensors studied.

### 5.1. The Hybrid FS-DT Model Applied for Predicting PM1 Concentrations

After conducting a thorough review of the existing literature, it was observed that no papers were identified that focused on predicting and/or modeling PM1 concentrations. Additionally, the WHO recommendations did not provide AQI classifications specifically based on PM1 concentrations.

In response to this gap in research, our work employed hybrid FS-ML models to predict PM1 concentrations. This decision was motivated by the belief that PM1, despite being less explored, could adversely affect human health and the ecosystem.

In Figure 8, all possible combinations of meteorological variables to predict PM1 concentrations were examined across all studied stations/sensors. The results indicate a consistent pattern, with pressure emerging as the primary significant predictor for almost all sensors, except for sensor 16000209. In the case of sensor 16000209, temperature took precedence as the first significant predictor, followed by pressure and humidity. This divergence could be attributed to the geographical coordinates or climate characteristics unique to the location of sensor 16000209.

The results found here indicate that humidity has a lower influence on PM1 concentrations. Generally, the R^2^ was between 0.5 and 0.9, 0.7 and 1, and 0.4 and 0.7 for temperature, pressure, and humidity, respectively. The best accuracy was discovered by combining pressure with temperature and slightly with humidity. This accuracy is justified by the R^2^ correlation between 0.9 and 1 and by the indication of dispersion, the MAPE, and the σ being closer to zero.

Moreover, excluding sensors 1600020A and 16000238, the best accuracy was shown by combining pressure with temperature, while for other sensors, humidity was added to slightly enhance the prediction’s accuracy. The statistical results indicated almost perfect correlation and approximations between the measured values and the PM1 predicted by these two combinations. R^2^ was found to be closer to 1 and MAPE and σ to 0. Other results can be extracted from the same figure.

### 5.2. Hybrid FS-DT Model Applied for Predicting PM2.5 Concentrations

In most articles that have been read, the authors have tried to predict PM2.5 and/or PM10 concentrations based on various sets of meteorological variables and by employing several machine learning methods. In most cases, correlations between these objective functions and the meteorological variables studied in this study do not reach the confidence interval of 0.95 for R^2^. The readers are referred to the references summarized in Table 2 to obtain this information. The result found here in this study shows correlations closer to 1 (accuracy close to 100%) for almost all stations/sensors studied (see Figure 9).

### 5.3. Hybrid FS-DT Model Applied for Predicting PM10 Concentrations

The coarse particulate matter PM10, known as atmospheric particles with a diameter between 2.5 and 10 µm, has a broad negative impact on human health, mortality level, and illness, as well as on the environment and ecosystems [35]. Researchers worldwide have widely investigated the possible relationship between local meteorological patterns, PM10, and air pollution. Several ML models were employed to accurately predict PM10 using numerous meteorological inputs. This subsection is coming from this context.

Like the above subsections, in Figure 10, all possible combinations of the meteorological variables considered here are checked for predicting the PM10 concentrations at all studied stations/sensors. Like for PM1 and PM2.5 concentrations, pressure is the main significant predictor, followed by temperature and humidity, respectively. For the sensor 16000209, temperature is the first key predictor, followed by pressure and humidity. This is to say that humidity has a more minor influence on PM10 concentrations. In addition, except for the sensor 1600020F, the best accuracy for all other stations/sensors was observed by combining pressure with temperature and a little with humidity. For the sensor 1600020F, the best accuracy was only given by combining pressure with temperature. This may be because this sensor is the sole one that registered outstanding Hazardous AQI values (4473 values). These remarks suggest we perform another study on the possible relationship between AQI or PM concentrations and the predictor variables studied for each station/sensor and each AQI category (Good, Moderate, Unhealthy for sensitive groups, Unhealthy, Very Unhealthy, and Hazardous categories).

### 5.4. Influence of VOC, Noise, and CO_2_ on PM Concentrations

The sensor 820002C3 was the sole sensor that, plus the three meteorological variables, measured noises, CO_2_, and VOC. In this case, the number of possible combinations was 63, and in Figure 11, the variables, given these combinations, are shown.

The impact of these added variables on PM1, PM2.5, and PM10 was thoroughly examined, and the summarized results are presented in Figure 12. As depicted, several combinations exhibited a near-perfect correlation (R^2^ close to 1) for all particulate matter. The optimal combination identified for PM1 was **Comb44**, comprising Pressure, Humidity, CO2, and VOC. For PM2.5, the most effective combination was **Comb61**, involving Temperature, Pressure, NOISE, CO_2_, and VOC. Likewise, the best combination for PM10 was **Comb58**, which included Temperature, Pressure, Humidity, NOISE, and VOC.

Based on these findings, the conclusion was that, in addition to the three meteorological variables previously examined, NOISE, CO_2_, and VOC exerted minor influences on predicting PM concentrations. However, their inclusion can contribute to a slight improvement in prediction accuracy. This conclusion indicates that PM concentrations are not strongly related to these measured variables, and they should be combined with another predictor variable to enhance the prediction accuracy. Other remarks can be revealed from the same figure.

### 5.5. Modelling of PMs and AQI

The optimal combinations of variables for each PM, identified through this study and using the LSR method, led to the establishment of new relationships between PMs and the studied meteorological variables (refer to Table A1 in Appendix A). Furthermore, a novel interface was developed based on the study’s findings, as illustrated in Figure 13. This interface serves as a tool for predicting the PM concentrations and AQI for a given sensor/location, leveraging the meteorological variables investigated in the study. By utilizing this interface, it is possible to efficiently predict PM concentrations and subsequently determine the AQI using the most effective combination of predictor variables for each station/sensor.

## 6. Conclusions

The conclusions drawn from this study can be summarized as follows:

(1) By applying different ML models and using the LSR method, the PM concentrations and AQI were predicted with an excellent correlation and approximation. Here, the values of R^2^ can exceed, in general, 0.96, and, in most cases, can reach 0.99 for the twelve stations/sensors studied.

(2) Among all employed ML models, the FS-DT model proved to be the best model for predicting the PM concentrations with very high correlation and approximations.

(3) The humidity was the least significant variable in the PM concentrations, while the best accuracy was found by combining pressure with temperature.

(4) It was found that there were strong correlations between PM2.5 and PM10 (close to 0.99) and between PM1 and PM10 (R^2^ was between 0.89 and 0.98).

(5) With the approach methodology applied in this study, data-driven models offer the potential to achieve a correlation closer to 1 and a better approximation to real values. However, their performance is dependent on the availability of training and validating data.

(6) NOISE, CO_2_, and VOC exert minor influences on predicting PM concentrations, and they should be combined with another predictor variable to enhance the prediction accuracy. Noise reflects only the rhythm of the city. This indicates we cannot build relationships between PM10 concentrations and these measured data.

(7) The modelling in this study, which provides real-time inputs within the scope of the continuity of air pollution monitoring in any environment, is quite reliable as an early warning with complete accuracy.

(8) The findings of this study will inspire work in this area to validate these models by other sensors to predict PMs and other missing variables given by the sensors.

(9) For local communities, it is essential to find out the level of pollutants in the air, both from official and independent networks of sensors/stations, helping decision makers to develop programs and implement proper measures and regulations to reduce air pollution.

(10) To enhance the developed models’ performance, at least one of the other meteorological parameters (solar radiation, wind speed, direction, and cloudiness, etc.) should be considered in the optimization process and inserted in the modelling steps.

(11) For sensitive people, checking the air quality before deciding to spend time outside is helpful. Also, it is useful for tourists to know the air quality when choosing a vacation destination. The monitored system might relate to a datalogger device to detect high AQI values and set alerts. These alerts can be launched on a platform dedicated to air pollution or a mobile application for the public.

(12) Considering the future decline in air quality, modelling air pollution is important for everyone, because each life is impacted by air pollution. There are still unknown local factors that influence air pollution. In perspective, if more datasets are accessed simultaneously from Environmental National Agencies, independent sensor networks, and satellites (Copernicus Atmosphere Monitoring Service), the quality of the prediction will significantly increase, even if they use different measurement methods. The complementarity of the datasets is vital. Sources of air pollution will be identified more easily if sensor networks for air pollution monitoring are developed and the sensors have a higher density. Considering the complementarity of the data from different institutions that monitor air pollution might help to improve the quality of prediction in this field.

Finally, undoubtedly, the findings of this study will contribute to increasing the current level of knowledge on the prediction of air pollution and will add significant richness to the literature within the scope of studies in this field. In addition, the findings of this study might be an essential evaluation tool for decision making.

## Figures and Tables

**Figure 1 sensors-24-01532-f001:**
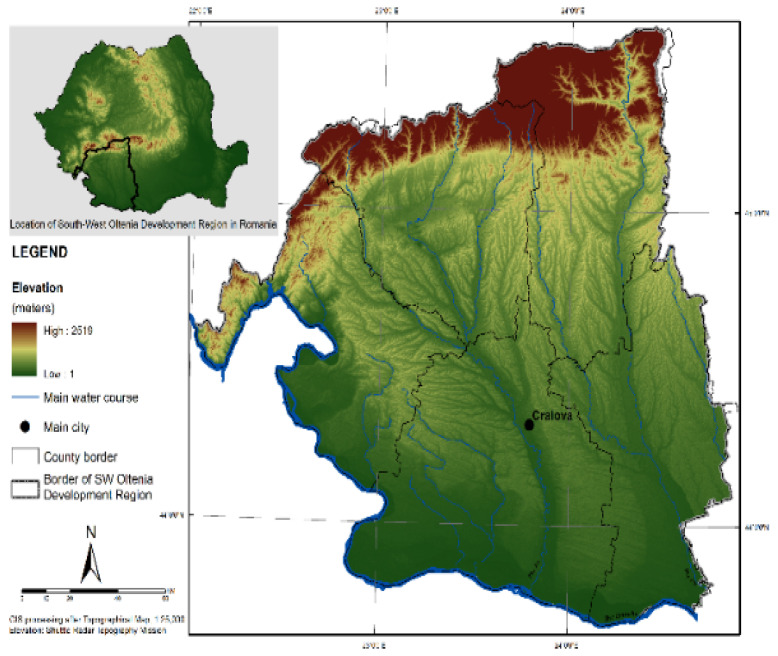
Craiova localization.

**Figure 2 sensors-24-01532-f002:**
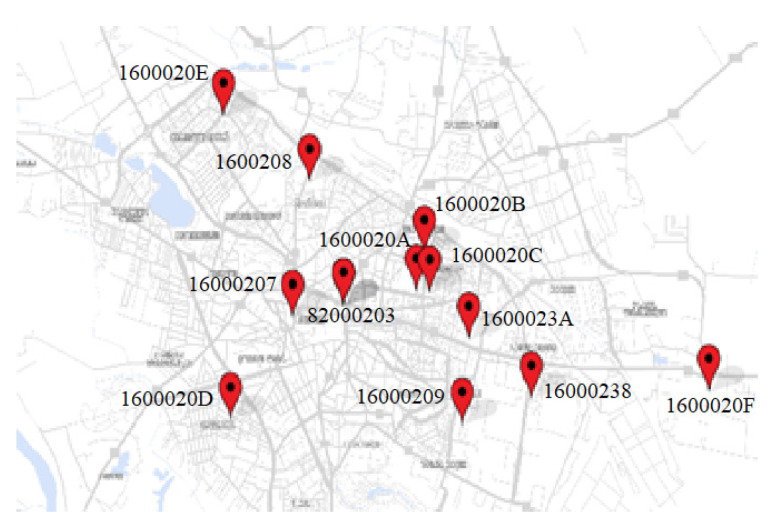
Distribution of the PM sensors in Craiova.

**Figure 3 sensors-24-01532-f003:**
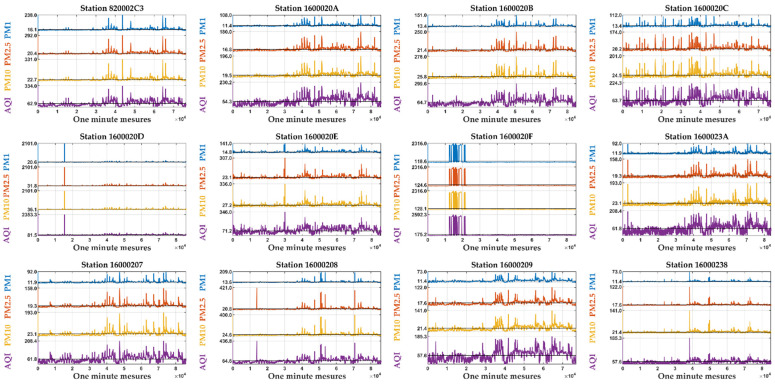
AQI distribution versus the PMs concentrations at all studied stations.

**Figure 4 sensors-24-01532-f004:**
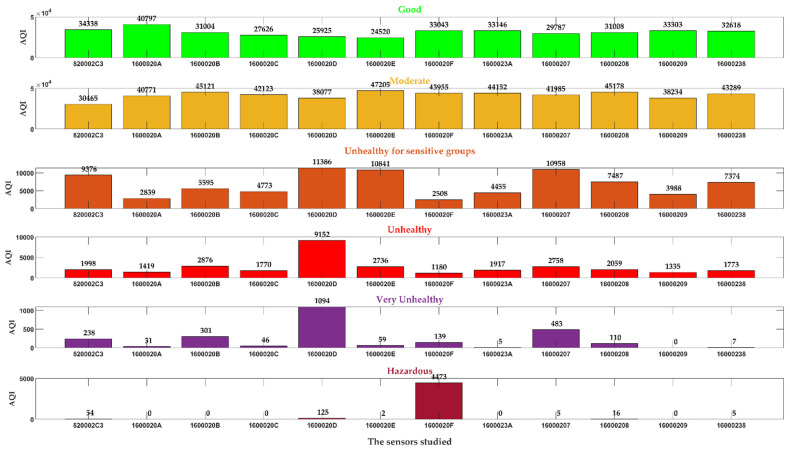
AQI clustering.

**Figure 5 sensors-24-01532-f005:**
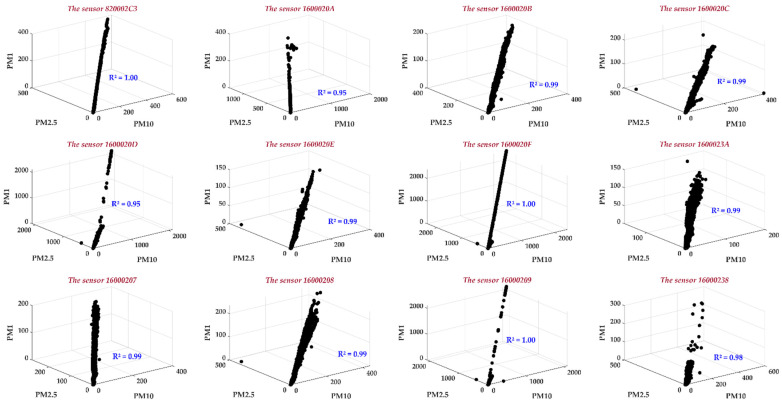
Correlation between PM10, PM2.5, and PM1 for all studied sensors.

**Figure 7 sensors-24-01532-f007:**
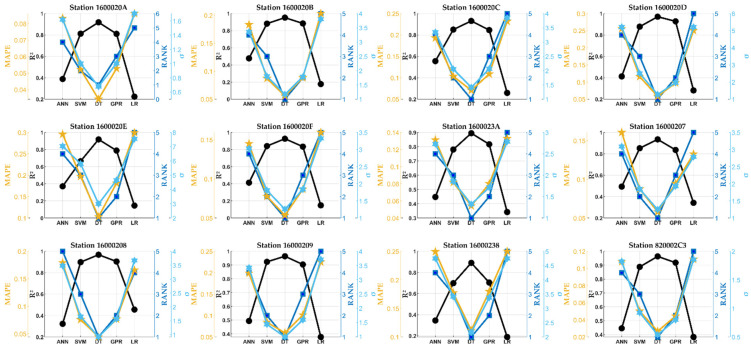
A statistical comparison of the five studied ML models.

**Figure 8 sensors-24-01532-f008:**
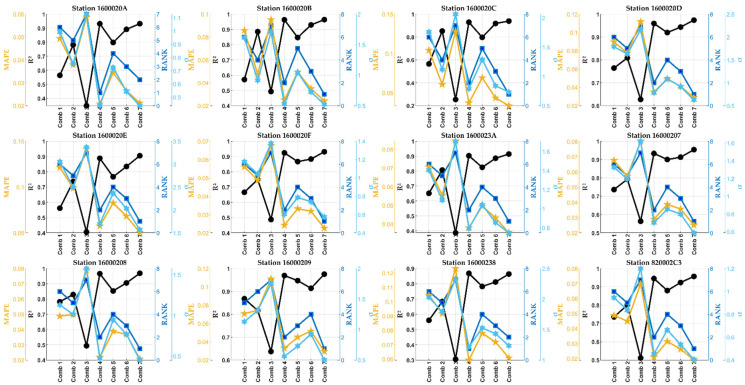
The results were found by applying the hybrid FS-DT model to the PM1 concentrations for all studied stations/sensors.

**Figure 9 sensors-24-01532-f009:**
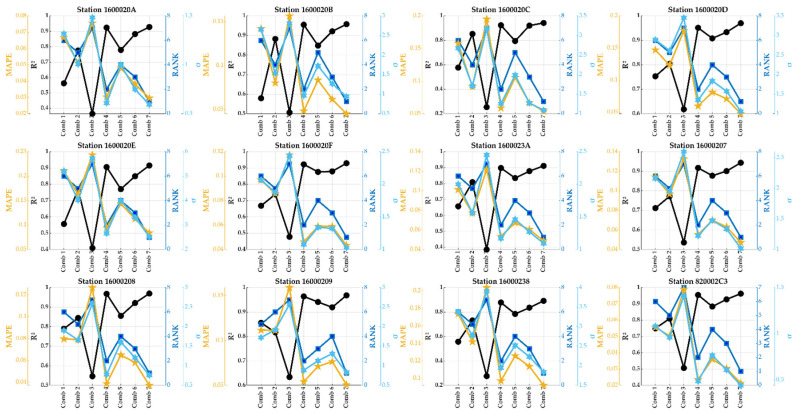
The results were found by applying the hybrid FS-DT model to the PM2.5 concentrations for all studied stations/sensors.

**Figure 10 sensors-24-01532-f010:**
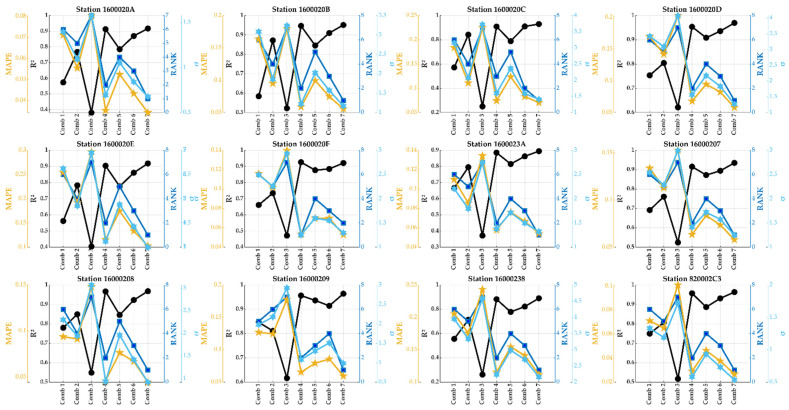
The results were found by applying the hybrid FS-DT model to the PM10 distribution for all studied stations/sensors.

**Figure 11 sensors-24-01532-f011:**
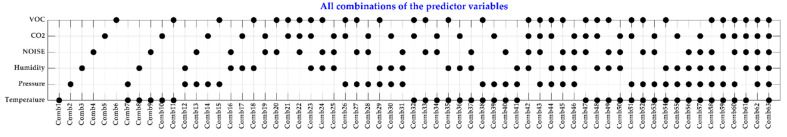
The predictor variables involved in total combinations of inputs.

**Figure 12 sensors-24-01532-f012:**
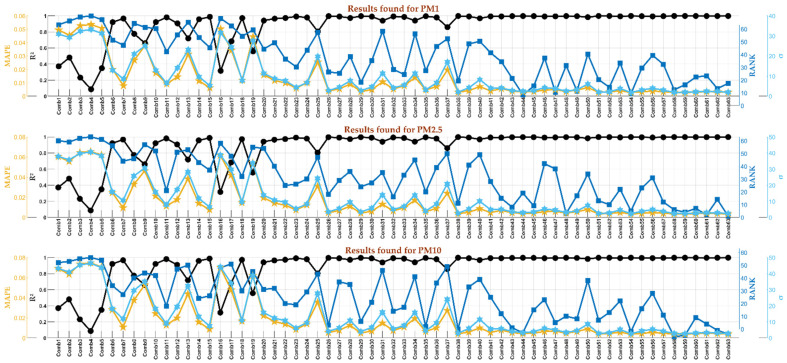
The results were found by applying the DT model to the noise and CO_2_ versus PM10.

**Figure 13 sensors-24-01532-f013:**
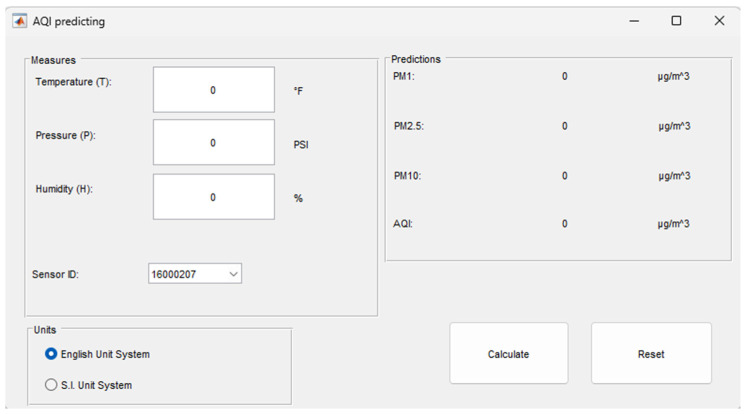
The AQI prediction interface is elaborated within the study.

**Table 1 sensors-24-01532-t001:** Abbreviations and nomenclature.

Abbreviation	Nomenclature	Units
ANNs	Artificial Neural Networks	--
LCE	Legate’s Coefficient of Efficiency	Dimensionless
LSR	Least Square Regression	--
MAPE	Mean Absolute Percentage Error	In percentage
MBE	Mean Bias Error	μg/m^3^
EWT	Ensemble Wavelet Transform	--
VMD	Variational Mode Decomposition	--
NARX	Network nonlinear Autoregressive Network with Exogenous Inputs	--
ARIMA	Auto Regressive Integrated Moving Average Model	--
MAEGA	Multi-Agent Evolutionary Genetic Algorithm	--
ELM	General Neural Networks	--
LSTM	Deep Learning Neural Networks	--
MODA	Multi-objective Dragonfly Optimization Algorithm	--
MOPSO	Multi-objective Article Swarm Optimization Algorithm	--
MOBO	Multi-objective Bonobo Optimizer	--
PM	Particle Matter Concentration	μg/m^3^
R^2^	Coefficient of Determination	Dimensionless
ML	Machine Learning	--
RH	Relative Humidity	In percentage
P	Pressure	Pa
RMSE	Root Mean Square Error	μg/m^3^
SBF	Slope of Best-Fit line	Dimensionless
FS	Feature Selections	--
T	Temperature	°C
TS	Test Statistic	Dimensionless
WIA	Willmott’s Index of Agreement	Dimensionless
σ	Standard Deviation	μg/m^3^
φ	Performance Score	Dimensionless
MDA	Mixture Discriminant Analysis	--
Bagged CART	Bagged Classification and Regression Trees	--
RF	Random Forest	--
SA	Simulated Annealing Method	--
SVM	Support Vector Machine	--
DT	Decision Tree	--
GPR	Gaussian Process Regression	--
LR	Linear Regression	--
RF-ELM	Random Fourier Extreme Learning Machine	--
RF-ELM	Random Fourier Extreme Learning Machine	--
OS-ELM	Online Sequential Extreme Learning Machine	--
IVS	Input Variable Selection	--

**Table 3 sensors-24-01532-t003:** AQI significance in terms of health.

AQI	Air Quality Conditions for Health
0–50	Good
51–100	Moderate
101–150	Unhealthy for sensitive groups
151–200	Unhealthy
201–300	Very unhealthy
301–500	Hazardous

**Table 4 sensors-24-01532-t004:** The considered inputs and output parameters.

Input and Output Number	Parameter	Unit
Input 1	Temperature	°C
Input 2	Pressure	Pa
Input 3	Relative Humidity	%
Input 4	NOISE	----
Input 5	CO_2_	μg/m^3^
Input 6	VOC	----
Output 1	PM1	μg/m^3^
Output 2	PM2.5	μg/m^3^
Output 3	PM10	μg/m^3^

## Data Availability

(a) The dataset, models, or codes supporting this study’s findings are available from the corresponding author upon reasonable request. (b) All data, models, and code generated or used during the study appear in the submitted article.

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
