# Peer review of "Multivariable Air-Quality Prediction and Modelling via Hybrid Machine Learning: A Case Study for Craiova, Romania"

_sensors, 2024, doi:10.3390/s24051532_

Round 1
Reviewer 1 Report
Comments and Suggestions for Authors
1 Overall evaluation
(1) Main contribution: The structure of this paper is clear. The data set measured by 12 sensors (temperature, humidity, and air pressure) is used for choosing optimal item of prediction accuracy among 5 machine learning methods, and the method for predicting PM1 concentration is provided, with high accuracy for predicting PM2.5 concentration. At the same time, the correlation between the predicted PM10 concentration and noise and CO2 emission is preliminarily excluded. The influence of temperature, humidity and air pressure on the prediction accuracy is also discussed.
(2) Defects of the paper: the details need to be improved, and some mistakes could have been avoided; The data analysis part is lacking and can be more in-depth; Limited innovation, the paper is to use existing machine learning methods to solve the problem of air quality prediction.
2 Specific modification suggestion
2.1 Major comments
(1) About data sampling: It is necessary to give a summary table of the characteristics of different sensors, because there are many special cases of sensor data in the article. If it is the difference of data sets, hope that author can discuss the difference and characteristics of different data sets in data analysis. In line 181, the sensor naming sequence number is somewhat disordered, and its meaning needs to be further explained. If there are special points in the setting of different sensors, it should also be explained in advance. In line 230, since there will be some loss of detection data, why not change the sensor type? Line 426, why is the prediction accuracy of the combination also affected by the sensor, shouldn't the sensor only serve the purpose of collecting data? Line 468, why is the most accurate prediction combination of the 1600020F sensor special, and do other sensors have such special cases?
(2) About data analysis: the part of data and data analysis mainly focus on the display of data, and it is necessary to enrich the data analysis. For example, the data in Figure 3 and 4 can be properly analyzed for reasons.
(3) About data training and testing: In Figure 11, it is necessary to input the device ID when entering data, does it mean that each sensor data is trained as an independent data set? If the data from each sensor is used as an independent set of predictions, can the data be combined for training? Line 492, in the actual air quality prediction process, does the interface program automatically determine the optimal combination of inputs, or does the optimal combination summarized in the previous study need to be used?
2.2 Minor comments
(1) In line 18, it is mentioned in the abstract that the relationship between PM concentration and noise, carbon dioxide emission and other variables is studied. In fact, only the predicted relationship between PM10 and noise and carbon dioxide emission is discussed in the paper, and it is believed that the expression in the abstract can be more rigorous.
(2) Line 229, date is wrong, 2011 should be changed to 2021.
(3) Line 305, title 3.5 error, should be Linear Regression.
(4) In line 396, after selecting the DT method, it is recommended to explain the meaning of the FS-DT method, and the explanation of FS in Table 1 is not very clear.
(5) In line 442, the expression of ‘this study’ is ambiguous, and it is suggested to add a transition word before it.
(6) Figure 1 lacks clarity.
(7) Figure 3, the data on the vertical axis overlaps, so it is recommended to adjust it.
(8) The representation in Figure 4 is not clear enough, and the amount of data itself is small, so it is better to replace it with a table.
Author Response
Thank you for the constructive review of our manuscript “Multivariable Air-Quality Predicting and Modelling via Hybrid Machine Learning; A Case Study for Craiova, Romania.” Those comments were very helpful in revising and improving the quality of this paper. We have tried our best to improve the manuscript carefully, following your comments and suggestions. Our point-by-point response to the comments is given in the attached document.

Reviewer 2 Report
Comments and Suggestions for Authors
The conclusions refer to many aspects, but in reality are not conclusive. Even the authors recognize some of them as first step to be further studied, or as no conclusive/good/recommended correlation.
The data base of the input values are not from own measurements, it is based on several sensors positioned in the city, but still not correct described.
The authors mention:
The twelve sensors are PM Smoggie model, and only one is an 191 A3 model (820002C3). Sensors are produced by a Romanian start-up focused on innovation and are calibrated by the manufacturer. Three international independent laboratories 193 stated that PM Smoggie and A3 sensors under evaluate the PM concentrations.
Under these conditions what credibility arrises?
Very important is if the method used in the manufactured sensors are standard, according EU regulations, and if the instruments are calibrated. For sure yearly calibration is lacking. So please comment on these ideas and bring proofs for the statements.
Yes it is important to predict air quality, but the given model and example is not of best choice.
After indicating some formulas and calculation of the AQI (based on other several pollutants in addition to PM) for me it is not clear why you have selected noise as correlation actor with PM's? at lease give an explanation/reasons for your decision
In the Abstract there are to many ideas, but not very clear the scope focused.
No quality control is presented. Errors?
Literature seams a little bit older.
Author Response

(The authors gave the same response as above.)

Reviewer 3 Report
Comments and Suggestions for Authors
This paper presents a case study on air quality prediction and modeling in Craiova, Romania using a holistic approach and hybrid machine learning models. The study investigates the associations between meteorological factors and particulate matter (PM) concentrations, as well as the correlation between PM concentrations and other variables. Five hybrid machine learning models are used to predict PM concentrations and calculate the Air Quality Index (AQI). The results show high prediction accuracy for the models, with coefficient of determination (R²) values exceeding 0.96. However, this study needs to address the following concerns before it gets published:
Formatting:
1. "6.735" in Line 60 should be "6,735", and the authors are suggested to give a citation for the World Quality Report as well.
2. Considering that the authors used a subscripted format for the numbers in the chemical formulas describing pollutants such as NO2, then the authors are also suggested to use a subscripted format for the numbers in the PM-related chemical formulas.
3. The title of subsection 3.5 on Line 305 should read: "Linear Regression".
4. Line 351 heading number should read "5." not "5.5".
5. Is the sentence "and the worst sub-index (the max value) communicates the AQI)" in Lines 482 to 483 open?
Content:
1. Lines 193 to 194 mention "Three international independent laboratories stated that PM Smoggie and A3 sensors under evaluate the PM concentrations.". "What are these three independent laboratories? What were the details of the comparison experiment? What was the standard measurement equipment used for the comparison? The authors are suggested to add specifics, which are critical for quality control of sensor monitoring data. The current description of the monitoring sensors in the manuscript is too short to demonstrate that the accuracy of the sensors' monitoring can meet the needs of scientific research.
2. What exactly is the Autonomous Anomaly Detection method mentioned in Line 201? How does this method achieve the detection of anomalous data? Could the author please specify this method?
3. In Line 210, AQI is an index used to assess the goodness of air quality and it is a dimensionless parameter. Then the author's AQI with an unit in Table 3 should refer to the specific concentration of PM pollutants, and here the author may have confused the definition of AQI. The authors are suggested to check the calculation method of AQI and the conversion tables developed by different regional environmental protection departments to do the correct calculation of the AQI parameter.
4. The authors specifically describe the five hybrid FS-ML models used in this study in Section 3 of the manuscript, but Sections 3.6 and 3.7 describe some of the data selection methods they used in comparing the five algorithms. It is recommended that the authors separate these two sections into a separate section, otherwise mixing them with the presentation of the FS-ML models will be more confusing to the reader.
5. Are the combinations of input variables consistent for the best model for each site in Figure 6? If not consistent, what are they? Please clearly label them in the figure.
6. The authors have just started to justify the accuracy of the DT model in predicting pollutant concentrations based on meteorological information in Section 5, but what is the rationale for the direct use of the DT model in the study of PM1.0 and PM2.5 concentrations for the prediction of PM10 concentrations in Section 5.1? Considering the large differences between the input variables of the two parts of the study, the authors should have given more justification in terms of model selection in section 5.1.
7. What are the details of each of the 7 input combinations in Section 5.6? Please list them specifically. In addition, why did the authors not choose to use real-time monitored concentrations of noise and CO2 as input variables?
8. The discussion of the results could be clearer and more focused on the implications and potential applications of the findings. Therefore, could the authors provide more insights into the potential practical applications of your findings in improving air quality management in urban areas?
Comments on the Quality of English LanguageThe authors need to polish the English writing of this manuscript because there are some obvious grammar errors.
Author Response
We sincerely appreciate all valuable comments and suggestions, which helped us improve the article's quality. In the following, we highlight the editor's concerns and our efforts to address these concerns. In response to the editor’s constructive criticisms, we have also made numerous revisions, which -we believe- improved the manuscript's quality.

Round 2
Reviewer 2 Report
Comments and Suggestions for Authors
The links for validation of the sensors are not very helpful
One of these indicates for example
PM10 mass concentration measurements measured by uRADMonitor A3 sensors do not correlate with the corresponding GRIMM, FEM BAM and T640 (R2 ~ 0.15 , 0.20 and 0.38, respectively, 1-hr mean) and underestimate PM10 mass concentration measured by the reference instruments
• No sensor calibration was performed by SCAQMD Staff prior to the beginning of this test
• Laboratory chamber testing is necessary to fully evaluate the performance of these sensors under known aerosol concentrations and controlled temperature and relative humidity conditions
• All results are still preliminary
The final version of the article is not easy to be read, even it is colored (new parts introduced) and proves also the cut parts. I expect another one - in final form. Make use of the validation of the sensors, this is very important. GRIMM instrument is not an EU standard measuring method.
THE NEW CONCLUSIONS ARE NOT LINKED TO THE SCOPE OF THE ARTICLE AND ITS TITLE
Please keep in mind that measuring air pollution must be clear and if evaluated, the prediction must be indicated with an error coefficient
The article has a lot of *personal addressing* aspects, avoid *we, our.. etc*. It is a scientific report not literature.
The article could be very good with more involvement, I presume. Check English as well if possible.
Author Response
We sincerely appreciate all valuable comments and suggestions, which helped us improve the article's quality. In the following, we highlight the reviewer's concerns and our efforts to address these concerns. In response to the reviewers’ constructive criticisms, we have revised the manuscript, which -we believe- improved the manuscript's readability, which was the common concern of the reviewer.

Reviewer 3 Report
Comments and Suggestions for Authors
The revised manuscript corrects the previous problems, while the authors focused on significant revisions to the results section, which is Section 5. Therefore, I recommend the paper for publication. But before the paper is published, the author still needs to provide additional explanation in the article on one issue:
1. Are the calculations of the different model prediction outcome metrics in Figure 7 based on the average of the seven combinations of meteorological parameter inputs, or the one meteorological parameter combination that achieves the best metrics for each model at each station?
Author Response
Thank you for your inquiry. Figure 7 assesses the prediction capabilities of each machine learning model in predicting PM10, and the optimal model for each station is selected. The analysis is conducted using a single combination of inputs that includes all predictor variables.